# Bioclimatic Modelling Identifies Suitable Habitat for the Establishment of the Invasive European Paper Wasp (Hymenoptera: Vespidae) across the Southern Hemisphere

**DOI:** 10.3390/insects11110784

**Published:** 2020-11-11

**Authors:** Matthew W. F. Howse, John Haywood, Philip J. Lester

**Affiliations:** 1School of Biological Sciences, Victoria University of Wellington, Wellington 6140, New Zealand; phil.lester@vuw.ac.nz; 2School of Mathematics and Statistics, Victoria University of Wellington, Wellington 6140, New Zealand; john.haywood@vuw.ac.nz

**Keywords:** BIOCLIM, invasive species, MAXENT, *Polistes dominula*, species distribution model

## Abstract

**Simple Summary:**

The European paper wasp, *Polistes dominula* Christ (Hymenoptera: Vespidae), has become an invasive species across the globe. This wasp can reach high population densities and this, combined with its predatory nature, makes this insect a potential threat to biodiversity. There has been a lot of research conducted on this species throughout the northern hemisphere; however, little is known about their distribution in the southern hemisphere. Our objective was to identify where, in the southern hemisphere, *P. dominula* could become established. Two species distribution modelling approaches were used to make these predictions. Based on these models, there are large areas across southern South America, South Africa, southern Australia, and much of New Zealand that are likely to be at risk of further invasion by this species. These findings can be used to inform biosecurity measures in regions deemed at risk of invasion by this globally important pest.

**Abstract:**

Species distribution models (SDMs) are tools used by ecologists to help predict the spread of invasive species. Information provided by these models can help direct conservation and biosecurity efforts by highlighting areas likely to contain species of interest. In this study, two models were created to investigate the potential range expansion of *Polistes dominula* Christ (Hymenoptera: Vespidae) in the southern hemisphere. This palearctic species has spread to invade North and South America, South Africa, Australia, and more recently New Zealand. Using the BIOCLIM and MAXENT modelling methods, regions that were suitable for *P. dominula* were identified based on climate data across four regions in the southern hemisphere. In South America areas of central Chile, eastern Argentina, parts of Uruguay, and southern Brazil were identified as climatically suitable for the establishment of *P. dominula.* Similarly, southern parts of South Africa and Australia were identified by the model to be suitable as well as much of the North Island and east of the South Island of New Zealand. Based on outputs from both models, significant range expansion by *P. dominula* is possible across its more southern invaded ranges.

## 1. Introduction

Species distribution models (SDMs) are becoming increasingly important in ecology, due to their ability to help predict the potential distributions of invasive organisms. These models bring together known species occurrence records and environmental data to provide users with an estimation of the conditions a species requires to survive. This information can be used to identify locations that could support populations of a particular species [1]. Species distribution models have been used to guide the creation of more effective reserves [2], to project impacts of climate change [3], and to predict the spread of invasive species [4].

*Polistes dominula* Christ (Hymenoptera: Vespidae) is a well-known and wide-ranging invasive social wasp species [5]. A palearctic species native to Europe, North Africa, and parts of Central Asia, *P. dominula* has spread to both North and South America, South Africa, Australia, and more recently, New Zealand (Figure 1) [5,6,7]. This species is largely predatory by nature [8]. Similar to its Vespid relatives, *P. dominula* is a frequent predator of Lepidoptera larvae [9,10,11,12] though it is thought that this predatory wasp has a more generalist diet than that of other related species [13,14]. *Polistes dominula* has also been shown to have several competitive advantages over closely related species. Studies in North America and South Africa have shown that *P. dominula* nests are more productive than other *Polistes* species, able to produce more offspring over a longer active season [15,16,17]. This high nest productivity has led *P. dominula* to reach large population densities in their invaded areas.

The invasion of *P. dominula* across the globe has been fairly well studied but not equally across affected regions. In the northern hemisphere *P. dominula* has famously invaded from the east to west coasts of the United States over the last 50 years [19]. *Polistes dominula* is known to have become established throughout the southern hemisphere but its spread in these regions has been critically understudied. This invasive species appears to have established in the southern hemisphere by the 1980s in Australia [20] and Chile [21,22]. It was found to have established in Argentina by 2003 [23], South Africa by 2008 [24], and in New Zealand by 2016 [7]. Except in South Africa, little research has been conducted on the status of these invasive populations and how they have affected local ecosystems. Many of these invaded areas contain native invertebrate communities that already face threats from other invasive species, habitat alteration, and climate change [25,26,27]. The establishment of *P. dominula* throughout more of these regions may add to these threats. Thus, predicting and preparing for future invasions or range expansions may help mitigate this effect.

Since their inception, there have been many changes and improvements to how SDMs are formed. One of the earliest and most widely used methods is BIOCLIM [28,29]. BIOCLIM is a profile method of species distribution modelling, whereby the algorithm determines the environmental similarity between a target species’ current range and other locations, using a percentile distribution of values [30]. The model will designate a location more suitable if the environmental values are closer to the median values of known occurrence sites. BIOCLIM is a presence-only method and so does not require known absence data. Another modelling method, known as MAXENT, can also be used with presence-only data, but there are some key differences from BIOCLIM. MAXENT is a machine learning method. The aim of MAXENT is to minimize the relative entropy between the probability density estimated from the occurrence data and the probability density estimated from the rest of the landscape [31]. The MAXENT method was developed more recently than the BIOCLIM approach, is also widely used, and is thought to be one of the better performing modelling methods [32,33,34].

A species distribution model would identify regions that are conducive to supporting populations of *P. dominula* and provide an insight into where these wasps are likely to spread. In this study two prediction models were produced, respectively using the BIOCLIM and MAXENT methods. Using global occurrence and climate data, these models were used to predict the bioclimatic suitability of four regions across the southern hemisphere for the establishment of *P. dominula*. These findings can inform future biosecurity and control plans for regions anticipated to be impacted by this invasive wasp.

## 2. Materials and Methods

*Polistes dominula* occurrence data was downloaded from Global Biodiversity Information Facility database (GBIF) [18] using the “gbif” function in the package dismo [35] in R version 4.0.2 [36]. A total of 20,616 records from 125 published datasets were downloaded. As GBIF data was pooled from a range of sources, from peer reviewed studies to citizen reports, data quality could vary. It has been shown, however, that a combination of data from citizen science and long term expert surveying can still produce robust distribution models [37].

A number of data-cleaning procedures were carried out to ensure the best quality data were used in creating the models. Following the procedures outlined by Hijmans and Elith [30], data were prepared by first removing data points with missing latitude or longitude values. Data were then assessed for the presence of duplicate coordinates which were removed to prevent pseudo replication. This assessment was accomplished using “duplicated”, a base function in R [36] that identifies records with identical coordinates to others, which were removed. Remaining data points were then cross checked against a simple world map to identify any coordinates that were located on water. These values were likely to be occurrences recorded with low resolution coordinates and had to be excluded.

A set of climate variables containing temperature and precipitation was used for modelling the distribution of *P. dominula*. Climate data were obtained in the form of 19 environmental layers, each representing a global bioclimatic variable at 2.5-min (5 km^2^) resolution (available from: https://worldclim.org/data/worldclim21.html). The WorldClim database is based on global weather station data from 1970 to 2000 and provides high resolution, global layers of monthly climate data [38]. These monthly data were used to create the 19 annual bioclimatic variable layers used in this study (Table 1) and in a variety of other distribution models [39,40,41,42,43].

Pseudoabsence points were created by selecting 10,000 random points from around the globe. This number was chosen to provide an appropriate ratio of presence to pseudoabsence points and maximize model reliability [44]. Climate data was applied to these pseudoabsence points as well as the cleaned occurrence data using the “extract” function in the raster package [45]. Presence data were randomly partitioned into test and training data using the “kfold” function, as recommended by [30] and used by [34]. Model predictions were made using the training data, which contained 7397 of the 9246 occurrence points. The predictions were then tested against the remaining 1849 occurrence points as well as 10,000 random pseudoabsence points.

Variable selection was performed using stepwise logistic regression to produce a subset of variables that would be used in the models (Table 1). The use of an automatic selection method was chosen to remove bias from the selection process. The procedure adds or removes variables from generalized linear models, one by one, checking the significance of all variables in the model each time. If a variable in the new model is deemed nonsignificant it is removed. Variable importance was measured by the Akaike Information Criterion (AIC). Important variables are added to the model while less relevant variables are removed. Using this analysis, the subset of variables that produced the model with the lowest AIC was chosen to perform the subsequent predictions. This process removed unnecessary variables from the model and reduced multicollinearity. The generalized linear model used all presence and background data with associated climatic variables and assumed a binomial distribution for the occurrence of *P. dominula* at each global location.

Two prediction models describing the global distribution of *P. dominula* were made using the worldwide occurrence of *P. dominula* and their associated climatic variables. One model was built using the BIOCLIM modelling method while the other used the MAXENT approach with the default settings [30]. With these models, suitable climatic conditions were predicted and identified across four regions of the southern hemisphere. All these regions have been previously invaded by *P. dominula* and include southern South America, South Africa, Australia, and New Zealand. The models’ raw predictive outputs produced maps at 5 km^2^ resolution with each cell containing values of habitat suitability. For both models, this predictive value was a number between zero and one. Following the BIOCLIM method, a cell would have a value of one if the environmental variables were equal to the median value for the occurrence data [30]. A zero value is conversely applied to any cell that possesses climatic values lower than the 10th and higher than the 90th percentile values in the occurrence data. Raw output values produced by the MAXENT modelling method are an approximate probability that the species will be present, given the local environmental conditions, otherwise known as the relative occurrence rate (ROR) [31,46]. Using the “var.importance” function from the package ENMeval [47], values of variable permutation importance to the model produced by MAXENT were identified. Variable permutation importance is a percentage value showing how heavily the model depends on a specific variable. Values of each variable in training, testing and background data are permuted and the resulting model is evaluated. The degree to which the permuted data weakens the MAXENT produced model’s performance is normalised to a percentage for each variable and presented as variable permutation importance (Table 2) [34].

Presence/absence predictions show cells that contain predicted values of climate similarity above a determined threshold specific to each model. This threshold was determined as the maximum of the sum of the sensitivity (true positive rate) and specificity (true negative rate), also known as the maxSSS method [48]. Any cell with a value of climatic similarity over the threshold was predicted as a presence point, while any cell under this would be considered an absence [30]. This method of threshold selection has been recommended as appropriate when working with presence-only data [48,49].

The models were evaluated looking at the area under the receiver operating characteristic curve (AUROC, abbreviated to AUC) value. Test data with known presence or pseudoabsence status were entered into the models. The degree to which the model could correctly assign these data points to presence or pseudoabsence classes was used to calculate the AUC. This value is a number between zero and one and represents how accurately a model predicts presence/absence. A model with an AUC value of 1 predicts presence/absence with 100% accuracy while a model with an AUC score of 0.5 is one that predicts a presence or absence correctly 50% of the time [30,50].

To investigate patterns presented in the final models, histograms of the known occurrence climatic values were plotted with the “hist” function [36] and visually compared with the corresponding bioclimatic layer for each of the focus regions. This was conducted for the two bioclimatic variables with the highest permutation importance.

## 3. Results

After the data cleaning procedure, a total of 9246 occurrence points of *P. dominula* remained for use in the model. Of these points, 3028 occurred in an invaded range while the other 6218 points originate in assumed native range (Figure 1).

Stepwise logistic regression analysis using the 19 bioclimatic explanatory variables identified a subset of variables that produced the best model, as indicated by the lowest AIC value (Table 1). The analysis used generalized linear models predicting presence or absence of *P. dominula*, with variables added or removed based on changes in the resulting AIC values. The final model contained 15 of the 19 WorldClim variables with the analysis removing Minimum Temperature of the Coldest Month (Bio6), Annual Precipitation (Bio12), Precipitation Seasonality (Bio15) and Precipitation of Driest Quarter (Bio17) (Table 1).

Modelling produced by the BIOCLIM method showed a number of regions in southern South America to have a climate conducive to the establishment of *P. dominula* (Figure 2). Once the threshold was applied, the model identified areas of central Chile, central and eastern Argentina as well as parts of Uruguay and southern Brazil as climatically suitable (Figure 3). Records of *P. dominula* are currently restricted to Chile and western Argentina, indicating the potential for a range expansion eastward. The raw output of the model produced by MAXENT highlighted a wider area of potentially suitable habitat than that of BIOCLIM (Figure 2); however, once the threshold was applied a more conservative potential range was predicted (Figure 3). Though not as expansive as the range predicted by BIOCLIM, it follows a similar pattern. Two main clusters of suitable habitats were identified by the MAXENT method with one spreading throughout central Chile and another in eastern Argentina and southern Uruguay (Figure 3).

In southern Africa, the BIOCLIM method identified an extensive range of climatically suitable habitat. The raw output of the BIOCLIM method highlighted areas of South Africa from the southwest of the country, eastward into the interior, encompassing much of Lesotho and into parts of Eswatini (Figure 2). With the threshold applied, much of this described area was predicted to be suitable for *P. dominula* (Figure 3). The raw output of the MAXENT method highlighted a wider area of the region but followed a pattern much the same as that of the BIOCLIM prediction (Figure 2). With the threshold applied, however, the MAXENT model identified a much smaller potential range than that of BIOCLIM. Only areas in the southwest of South Africa were predicted to be climatically suitable for the establishment of *P. dominula* (Figure 3). These areas fell within the South Cape province, which contains all known *P. dominula* occurrences within the southern Africa region.

Both prediction models identified much of the southern portion of Australia as climatically suitable for the establishment of *P. dominula.* The raw output of the BIOCLIM method highlighted most of the country south of approximately −30 degrees latitude (Figure 2). Once the threshold was applied the model identified most of this area as climatically suitable, with most of southwest and southeast of the country denoted a potential present value (Figure 3). The raw output of the MAXENT method again highlighted a larger area of Australia than the BIOCLIM method, focused around the south and east of the country (Figure 2). When the threshold was applied the model identified two main clusters of suitable habitat in the south of Australia. One cluster was predicted around the southern parts of the state of Western Australia expanding eastward from where *P. dominula* is currently known to occur. The other cluster is focused around the south-east of the country in a pattern similar to, but more conservative than that of the BIOCLIM method (Figure 3).

In New Zealand, the raw output produced by the BIOCLIM method indicated suitable climatic conditions throughout the upper and lower parts of the North Island, with central and western areas deemed less suitable. Much of the eastern side of the South Island was highlighted by the prediction model as climatically suitable, while the west coast was not (Figure 2). Once the threshold was applied, much of the country highlighted in the raw output was predicted as climatically suitable for *P. dominula* (Figure 3). Similarly, the raw output produced by the MAXENT method highlighted an extensive area across New Zealand with much of the North Island and east of the South Island receiving the highest values of suitability (Figure 2). With the threshold applied, the MAXENT method predicted that much of the North Island and the east of the South Island is climatically suitable for *P. dominula* (Figure 3). The potential invadable range identified by both the BIOCLIM and MAXENT methods indicate that known populations of *P. dominula* in New Zealand are likely to expand their range.

The BIOCLIM method’s performance as evaluated by AUC was 0.970 while the MAXENT method’s performance was slightly higher with a value of 0.982 (Figure 4). These can be considered high AUC values as both values are close to 1, indicating the BIOCLIM and MAXENT methods were able to discriminate between the test-presence and background points 97% and 98.2% of the time respectively [50]. For the MAXENT method, variable permutation importance showed that annual mean temperature (Bio1) was the most important variable with a value of 37.9%, followed by precipitation of the coldest quarter (Bio19) at 15.9%, temperature seasonality (Bio4) at 11.4%, and mean temperature of the warmest quarter (Bio10) at 9.0% (Table 2).

Histograms were produced (Figure 5) for the highest-ranking climatic variables, as measured by permutation importance (Table 2). Known occurrences of *P. dominula* were found to experience average annual temperatures (Bio1) between −1 °C and 26.6 °C, with a sharp peak between 8 °C and 10 °C (Figure 5 and Figure 6). Known occurrences of *P. dominula* were found to experience values of precipitation during the coldest quarter (Bio19) of 0–1057 mm, with most falling between 100 mm and 300 mm (Figure 5 and Figure 7).

## 4. Discussion

*Polistes dominula* is a widespread invader with introduced populations extending throughout North America, parts of South America, South Africa, Australia, and, more recently, New Zealand [5,6,7]. This species has the potential to reach higher densities than other paper wasps [17], and their preference to live in close proximity to human habitation [51] makes that an issue for human health. Invasive wasps have been linked to declines of native invertebrate species across the regions studied here [52,53,54]. Populations of *P. dominula* have already established across the four regions in this study but the estimated models predict a range expansion that could result in this species becoming present throughout more of the southern hemisphere.

The prediction models based on the BIOCLIM and MAXENT methods share some similarities in their predicted range of *P. dominula* across the four regions. In Australia, predicted distributions of *P. dominula* produced by both models closely follow patterns of average annual temperature. The majority of *P. dominula* occurrences fell in regions where the annual average temperature (Bio1) is between 8 °C and 12 °C with most of the country north of the predicted range averaging over 15 °C (Figure 6). This same pattern can be seen in South America where the potential range of *P. dominula* appears to be constrained by cooler temperatures further south (Figure 3 and Figure 6). The models incorporate the contribution of other variables too; however, it is apparent that some variables influence *P. dominula* distribution more than others. For example, the east coast of the North and South Islands of New Zealand were identified as climatically suitable for the establishment of *P. dominula* while areas such as the West Coast region of the South Island appear to be much less suitable (Figure 3). The limitation on their distribution there is likely due to the high precipitation. Mean precipitation of the coldest quarter (Bio19) was shown to be the second most important variable contributing to the MAXENT model’s accuracy (Table 2). The mean value of Bio19 from all known occurrence data was 195 mm of precipitation. The West Coast receives over 500 mm of rain in the coldest quarter [55] and so is considerably wetter than most places known to contain *P. dominula*. Similar patterns are observed across the other regions with the west coast of Tasmania, Australia also receiving high precipitation, well above the levels preferred by *P. dominula* (Figure 3 and Figure 7).

The two distribution models did produce slightly different predictions. Both models had high AUC values (Figure 4) indicating robust models. The BIOCLIM method may be prone to overfitting when using many variables, leading to a narrower potential distribution [56] and that method can be prone to underpredicting potential distributions, possibly explaining the difference in raw model outputs (Figure 2) [40]. MAXENT, by contrast, has been repeatedly identified as a more reliable method, classed as a high performing, stable modelling approach compared to the BIOCLIM method [32,33]. Unlike the BIOCLIM modelling method, MAXENT weights variables differently depending on how their inclusion affects the models’ AUC. This fundamental difference in modelling strategy leads to differences in the model outputs and hence explains some of the differences between predictions.

Threshold selection is another area that may explain differences between the BIOCLIM and MAXENT predictions. Thresholds were chosen using the maxSSS method, necessarily producing a different threshold for each method. The BIOCLIM method fitted models with a comparatively low threshold of 0.005 and so ranked nearly all the cells highlighted in the raw output plot as a presence value. The threshold for the MAXENT method was set at a comparatively higher 0.38, meaning that only the most climatically suitable of the areas highlighted in the raw MAXENT output were denoted a presence value. This difference in threshold explains the pattern we see where MAXENT presence/absence predictions appear to be more conservative for Australia, South Africa, and South America (Figure 3). By contrast, in New Zealand the MAXENT method’s raw values of habitat suitability were so high that even with a more restrictive threshold, the areas denoted a presence status were larger than those resulting from the BIOCLIM method. Despite these differences in the final outputs both models predict areas that experience more mild average annual temperatures and are drier in the cooler months to be more suitable for the establishment of *P. dominula.* Both models predict a range expansion of *P. dominula* throughout the southern hemisphere.

*Polistes dominula* has been shown to be an important predator of invertebrates. They have been linked to the decline of invertebrates in invaded ranges around the world [9,57,58]. This is a pattern seen in invasive social wasp species due to their predatory behaviour and ability to reach high population densities [59,60]. With this species’ continued range expansion, it is likely that *P. dominula* will compound existing pressures to native invertebrate fauna across the southern hemisphere. Molecular diet analysis performed on related *Polistes* species, also invasive to New Zealand, showed that both native and introduced Lepidopteran species made up the largest portion of their diets [11,61]. Over half of all of New Zealand’s threatened native Lepidopterans are found on the east coast of the South Island [62], a region that was identified as climatically suitable for *P. dominula* by both models. The seasonal nature of the *P. dominula* lifecycle [17,63] means only those prey species that are abundant or breed during the summer months are likely to be targeted [52]. This seasonality will still likely impact assemblages of invertebrates in the predominantly temperate regions that are predicted to be most suitable for establishment of this species.

Both models predict rather significant areas of bioclimatically suitable habitat across the southern hemisphere; however, these models have only considered the effects of temperature and precipitation on *P. dominula* establishment. Observations from the field in both native and invaded ranges show that *P. dominula* appear to preferentially nest near human habitation [15,63]. This pattern of synanthropy is seen in related species [59] and has been shown to allow other taxa to invade regions where environmental conditions are unsuitable [64]. Additional modelling including human habitation and land-use as variables may be able to produce a more realistic prediction of *P. dominula* potential range. Physical barriers, potential corridors, long-distance dispersal events (natural or human-related), and climatic change may influence colonization patterns. *Polistes dominula* was the most commonly reported nuisance wasp species in Colorado only four years after it was first identified in the state [65]. This species’ ability to attain high densities close to human habitation could lead to the potential for an increased rate of human–wasp conflict in these predicted ranges.

Climate-based models such as the ones presented in this paper have been criticized for not considering biotic interactions when predicting habitat suitability [66,67,68]. Other studies consider the presence or absence of competitors [69,70,71]. This could be an important factor to consider in the spread of *P. dominula,* as other *Polistes* species are found in many of these regions. In southern South America there are 21 other species of *Polistes*, South Africa is home to 6 native species, Australia contains 15 other species while New Zealand contains two other invasive species of paper wasp [6]. These other, closely related species represent potential competitors that may impact the likelihood of establishment by *P. dominula* regardless of habitat suitability. While *P. dominula* has been shown to initially outcompete congener *Polistes* species, total displacement and replacement has not been shown to occur. It had been suggested that *P. dominula* was in the process of replacing the native *Polistes fuscatus* in the north-eastern United Sates [17,19,72]. A ten-year study of the two species in Michigan, USA, showed that despite initial displacement of *P. fuscatus* by *P. dominula,* populations eventually stabilized, likely due to the presence of a common parasitoid [73]. In South Africa, however, *P. dominula* continued to outnumber its native counterpart, *Polistes marginalis,* and maintain more productive nests despite the higher infection rates by a parasitoid [15]. Of the regions discussed in this paper, South America and Australia already contain a diverse range of *Polistes* species and so too presumably *Polistes* parasites and pathogens, which could have implications on invasion success.

## 5. Conclusions

Despite their relative simplicity, the use of climate based SDMs to predict potential ranges of species has been shown to provide strong predictive power [74]. Simple models such as these are still useful for making predictions over the broader scale and where more of the finer details of a species’ ecology are not well understood. This paper presents the first attempt to use species distribution models to identify potentially suitable habitat for the establishment of the invasive *P. dominula*. Information gathered from these SDMs can help inform governments and conservation groups about the likelihood of *P. dominula* establishing in their respective regions. Regions with no known populations of *P. dominula* that were highlighted as suitable may invest in early detection and control to prevent the species from becoming established. Methods used here could be used and built on in future work to predict the range of other invasive species.

## Figures and Tables

**Figure 1 insects-11-00784-f001:**
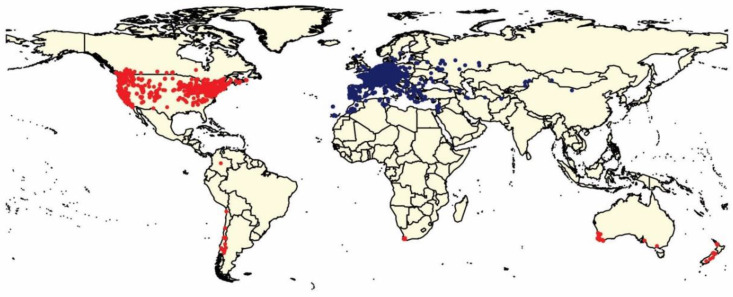
Global distribution of *P. dominula* constructed from data retrieved from Global Biodiversity Information Facility database (GBIF) [18]. Red points indicate occurrences in an invaded range. Blue points indicate occurrences in the assumed native range [5,6]. In total 9246 occurrences were used in this study. Of the total, 3028 occurrences were from invaded ranges and 6218 were from the assumed native range.

**Figure 2 insects-11-00784-f002:**
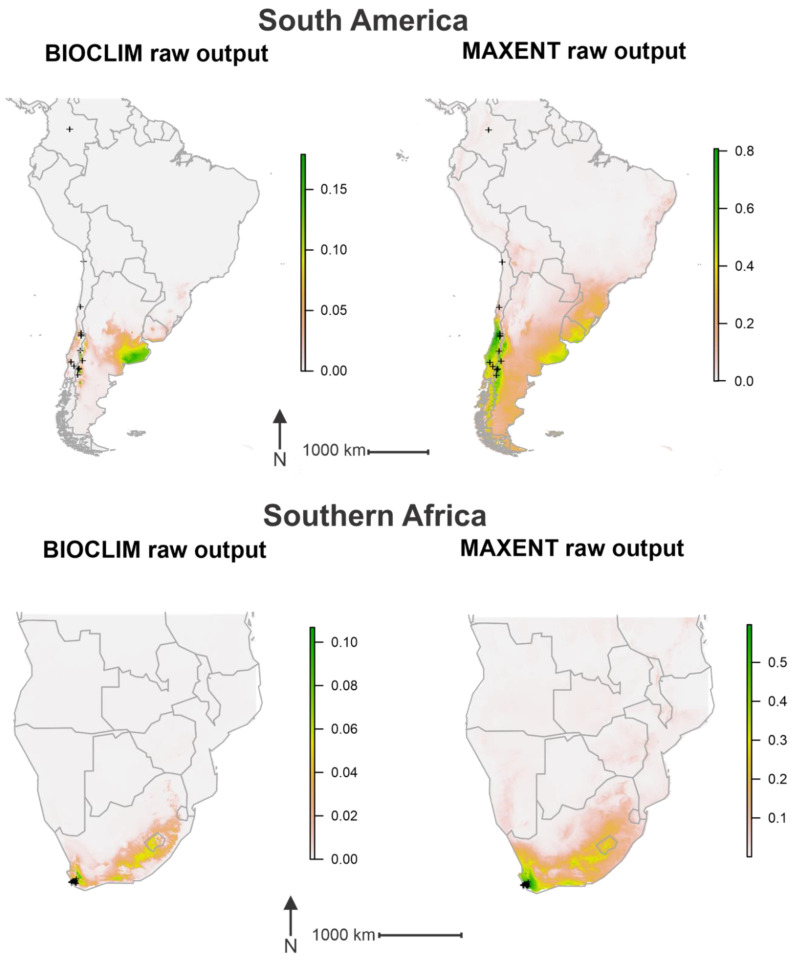
Raw outputs of both prediction models. The left images show the raw output given by BIOCLIM. The raw output of the BIOCLIM prediction is a value between 0 and 1 where the higher the number the more suitable the environment is. The BIOCLIM algorithm compares the environmental values of a cell to the median values of the environmental values of cells containing known occurrences of the target species. Percentile scores closest to 0.5 are most suitable so values over this are subtracted from 1. The resulting score is multiplied by 2 to get a final value between 0 and 1 [30]. The right images show the raw output given by MAXENT where the value is an approximate probability that the species will be present given the local environmental conditions [31,46]. The black crosses indicate known occurrences of *P. dominula*.

**Figure 3 insects-11-00784-f003:**
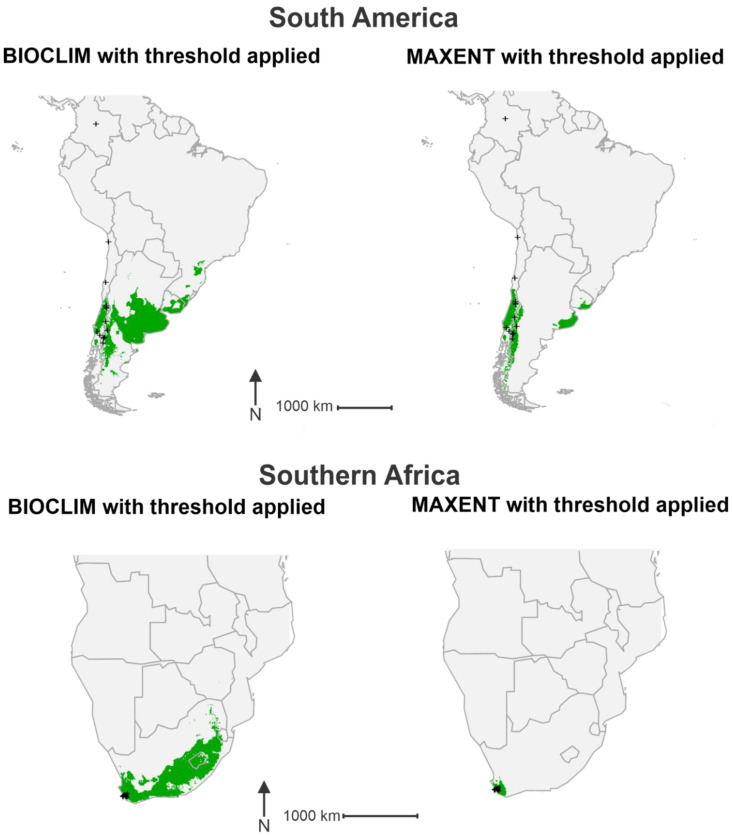
Predicted presence/absence plots for each region based on the two model outputs. For both models, thresholds (0.005 and 0.38, respectively) are calculated as the raw output value at which the sum of the true positive and true negative rates is maximized. Locations where the raw output values are over these thresholds are denoted a present status (1) and highlighted in green. Locations where raw output values are lower than the threshold are denoted an absent status (0) and remain grey. Black crosses indicate known occurrences of *P. dominula*.

**Figure 4 insects-11-00784-f004:**
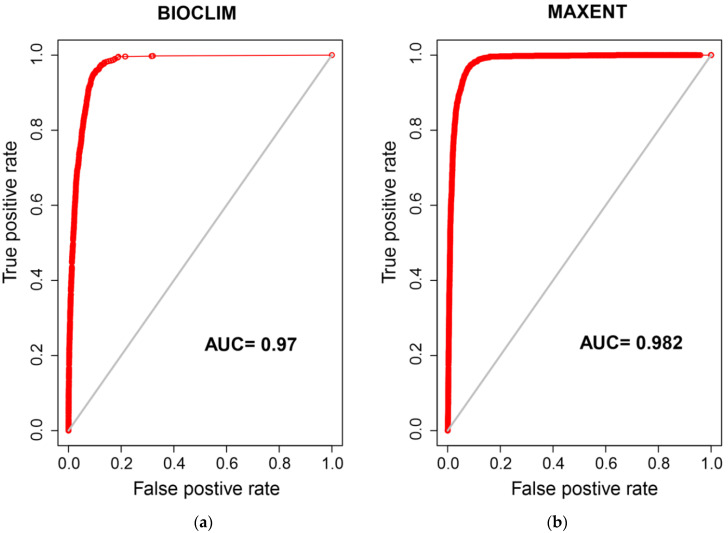
Area under the receiver operating characteristic curve (AUC) plots for BIOCLIM prediction (**a**) and MAXENT prediction (**b**). This value is a number between zero and one and represents how accurately a model predicts presence/absence. A model with an AUC value of 1 predicts presence/absence with 100% accuracy while a model with an AUC score of 0.5 is one that predicts presence or absence correctly 50% of the time [30].

**Figure 5 insects-11-00784-f005:**
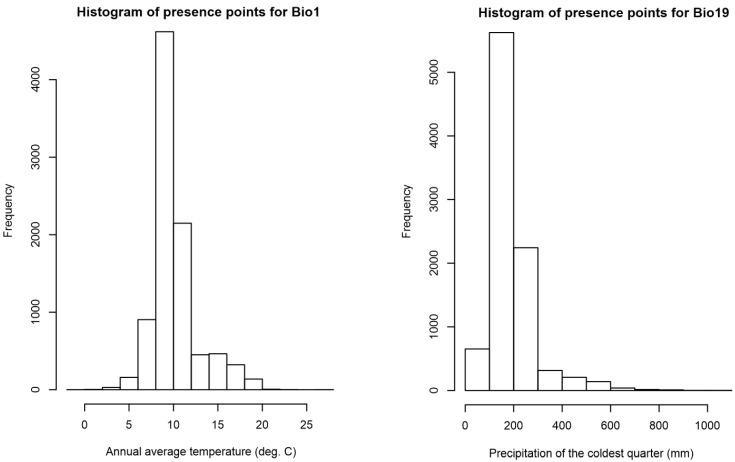
Histograms showing the distribution of bioclimatic values at known occurrences of *P. dominula.* Left is the distribution of annual average temperature (Bio1) and on the right the distribution of precipitation of the coldest quarter (Bio19). These two variables were chosen from the full list of 15, based on their high permutation importance in the MAXENT modelling approach.

**Figure 6 insects-11-00784-f006:**
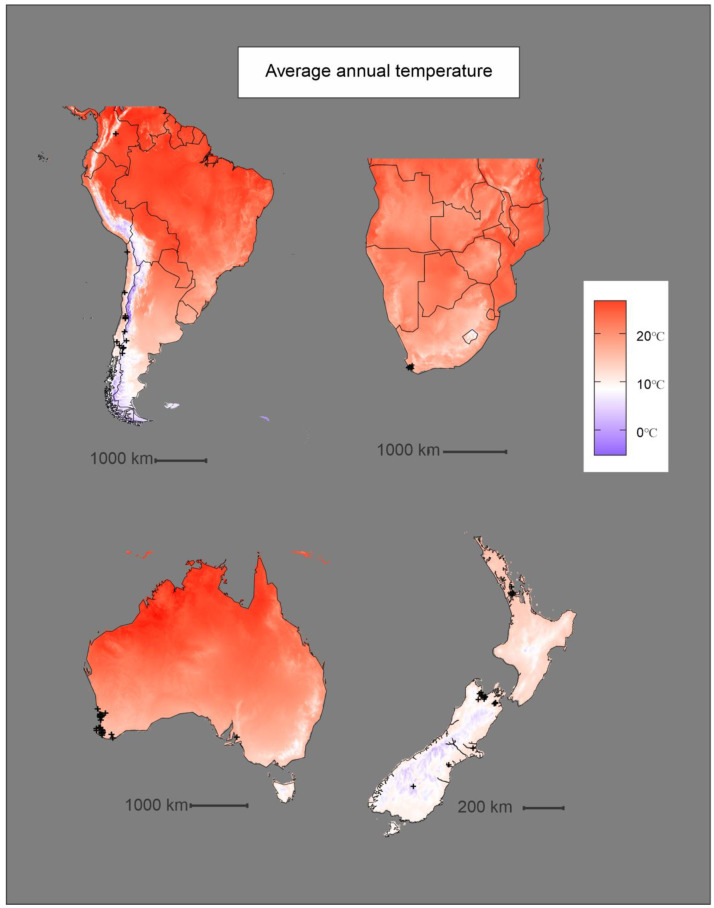
Plot of the average annual temperature (Bio1) bioclimatic layer at each region highlighted in this study. This variable had the highest permutation importance of 37.9%. Black crosses indicate known occurrences of *P. dominula.*

**Figure 7 insects-11-00784-f007:**
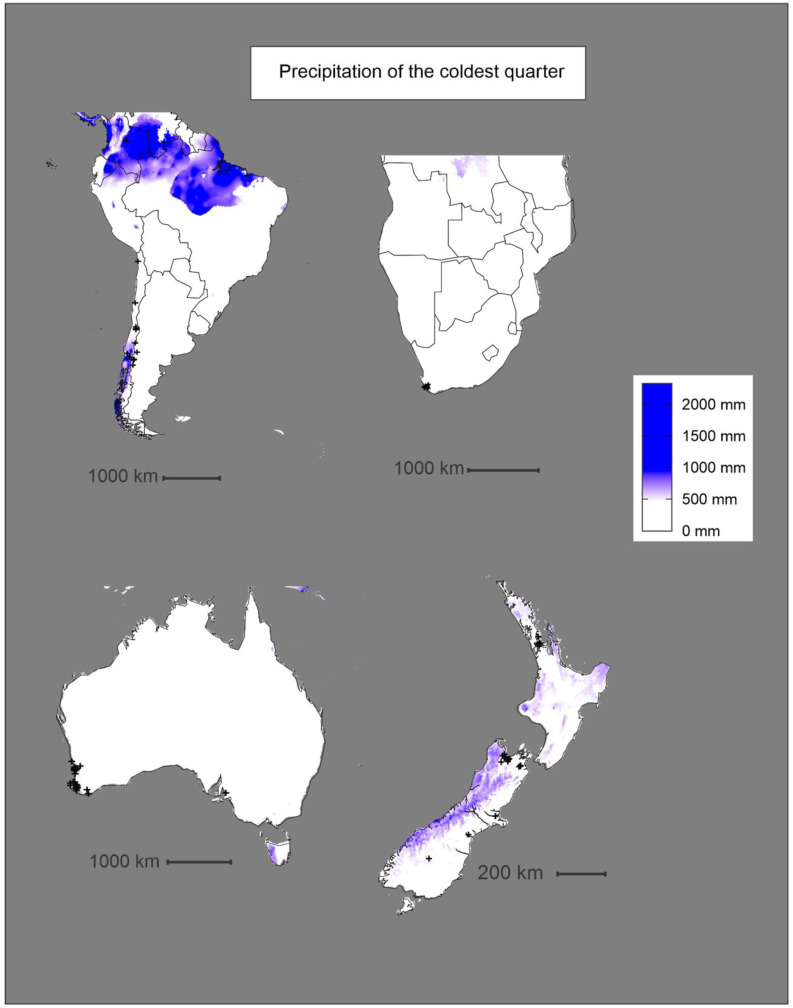
Plot of the precipitation of the coldest quarter (Bio19) bioclimatic layer at each region highlighted in this study. This variable had the second highest permutation importance of 15.9%. Black crosses indicate known occurrences of *P. dominula.*

**Table 1 insects-11-00784-t001:** List of bioclimatic variables used in this study. These variables, created by Fick and Hijmans (2017), were downloaded as 2.5 min (5 km^2^) resolution environmental layers from worldclim.org. Variables used monthly climate data collected over 30 years between 1970 and 2000. A subset of bioclimatic variables, identified by tick marks, was produced by stepwise regression analysis to be used in the BIOCLIM and MAXENT modelling approaches.

Variable Code	Variable Title	Unit	Inclusion in Final Model
Bio1	Annual Mean Temperature	°C	✓
Bio2	Mean Diurnal Range *(mean of monthly (max temp-min temp))*	°C	✓
Bio3	Isothermality *((Bio2/Bio7)* *× 100)*	%	✓
Bio4	Temperature Seasonality *(standard deviation* *× 100)*	°C	✓
Bio5	Max Temperature of Warmest Month	°C	✓
Bio6	Min Temperature of Coldest Month	°C	
Bio7	Temperature Annual Range *(Bio5-Bio6)*	°C	✓
Bio8	Mean Temperature of Wettest Quarter	°C	✓
Bio9	Mean Temperature of Driest Quarter	°C	✓
Bio10	Mean Temperature of Warmest Quarter	°C	✓
Bio11	Mean Temperature of Coldest Quarter	°C	✓
Bio12	Annual Precipitation	mm	
Bio13	Precipitation of Wettest Month	mm	✓
Bio14	Precipitation of Driest Month	mm	✓
Bio15	Precipitation Seasonality *(coefficient of variation)*	%	
Bio16	Precipitation of Wettest Quarter	mm	✓
Bio17	Precipitation of Driest Quarter	mm	
Bio18	Precipitation of Warmest Quarter	mm	✓
Bio19	Precipitation of Coldest Quarter	mm	✓

**Table 2 insects-11-00784-t002:** The 15 variables used in the MAXENT method, listed in order of permutation importance. Variable permutation importance is a value showing how heavily the final model depends on a certain variable. Values of each variable in training, testing and background data are randomized and the resulting model is evaluated. The degree to which the randomly permuted data weakens the model’s performance, as originally selected by MAXENT, is normalised to a percentage for each variable and presented as permutation importance [34]. A variable with a high permutation importance is therefore important to the model, since if its values were randomized, the power of the model would decrease significantly.

Variable Code	Variable Title	Permutation Importance
Bio1	Annual Mean Temperature	37.9%
Bio19	Precipitation of Coldest Quarter	15.9%
Bio4	Temperature Seasonality *(standard deviation × 100)*	11.4%
Bio10	Mean Temperature of Warmest Quarter	9.0%
Bio16	Precipitation of Wettest Quarter	4.7%
Bio18	Precipitation of Warmest Quarter	4.2%
Bio11	Mean Temperature of Coldest Quarter	3.8%
Bio3	Isothermality *((Bio2/Bio7) × 100)*	2.8%
Bio9	Mean Temperature of Driest Quarter	2.7%
Bio14	Precipitation of Driest Month	2.3%
Bio2	Mean Diurnal Range *(mean of monthly (max temp–min temp))*	1.8%
Bio5	Max Temperature of Warmest Month	1.4%
Bio13	Precipitation of Wettest Month	1.1%
Bio7	Temperature Annual Range *(Bio5-Bio6)*	0.5%
Bio8	Mean Temperature of Wettest Quarter	0.4%

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
