# Peer review of "Bioclimatic Modelling Identifies Suitable Habitat for the Establishment of the Invasive European Paper Wasp (Hymenoptera: Vespidae) across the Southern Hemisphere"

_insects, 2020, doi:10.3390/insects11110784_

Round 1

Reviewer 1 Report

The authors used two Species Distribution Models, BIOCLIM and MAXENT approaches, to assess the bioclimatic suitability of an invasive wasp species at a broad geographical scale. I found the study of interest and a good contribution to the knowledge of the bioecology of a wasp with potential adverse impacts on fauna and human health. Developing maps of key potential areas of establishment for invasive species is essential to decision making and successful management. The methods used are appropriate for the objectives of the work and, in general, well depicted. The resulting figures are sufficient, informative, and of good quality helping to follow the reasoning throughout the manuscript. In my opinion, the manuscript deserves publication after some minor changes mostly related to the terminology used throughout the text; please see the attached pdf file with comments/suggestions.

Reviewer 2 Report

Assessment of the paper entitled “The predicted distribution of the European paper wasp in the southern hemisphere (Hymenoptera; Vespidae)” for Insects (insects-988714)

Main comments

In this paper, the authors aimed to identify where, in the southern hemisphere, Polistes dominula – a wide-ranging invasive social wasp species – could become established. To do so, the authors used two species distribution modelling approaches to make these predictions (BIOCLIM and MAXENT). Therefore, both current species distribution data (from GBIF) and climate information were used to identify regions most climatically suitable for P. dominula invasion in the southern hemisphere. This line of investigation is very important and the results can be used to inform biosecurity measures in regions deemed at risk of invasion by this globally important insect pest.

I found this study well-written, clear and objective, with an interesting topic.  I also found that the use of two different modeling approaches brought important robustness to support the results obtained. The description of methods and the discussion of results are good. Therefore, I think this study will be a good contribution to the discussion on insect pest management across the globe.

Minor issues

Line 55. P. dominula is italicized.

Line 260. P. dominula is italicized.

Reviewer 3 Report

In general it is a well written paper with minor observations.      

Reviewer 4 Report

The submitted study by Howse et al. is very well written, the methods are appropriate without missing any key details. The conclusions are supported by the results obtained, recommending the use of climate based BIOCLIM and MAXENT models to predict the spread of the invasive Polistes wasp in the southern hemisphere. The work predicted significant range expansions across southern South America, South Africa, southern Australia and much of New Zealand. These findings should be used to inform biosecurity measures in regions deemed at risk of invasion by this wasp.

I enjoyed reading this manuscript and I could not find any egregious errors. Overall a nice piece of work.
